# Characterization of Gut Microbiota Composition in Type 2 Diabetes Patients: A Population-Based Study

**DOI:** 10.3390/ijerph192315913

**Published:** 2022-11-29

**Authors:** Isabella Polidori, Laura Marullo, Cristiano Ialongo, Flaminia Tomassetti, Roberto Colombo, Francesca di Gaudio, Graziella Calugi, Giulia Marrone, Annalisa Noce, Sergio Bernardini, Francesco Broccolo, Massimo Pieri

**Affiliations:** 1Lifebrain srl Cerba Healthcare, Guidonia Montecelio, 190/A Viale Roma, 00012 Rome, Italy; 2Lifebrain Nocera Cerba Healthcare, 84014 Nocera Inferiore, Italy; 3Department of Experimental Medicine, University of Rome “La Sapienza”, 00161 Rome, Italy; 4Department of Experimental Medicine, University of Rome “Tor Vergata”, 00133 Rome, Italy; 5Department of Medical Surgery, University of Palermo, 90127 Palermo, Italy; 6UOC of Internal Medicine-Center of Hypertension and Nephrology Unit, Department of Systems Medicine, University of Rome Tor Vergata, 00133 Rome, Italy; 7Department of Laboratory Medicine, Tor Vergata University Hospital, 00133 Rome, Italy; 8Cerba HealthCare Italia, 20137 Milan, Italy; 9Department of Medicine and Surgery, University of Milan-Bicocca, 20854 Milan, Italy

**Keywords:** dysbiosis, microbiota, diabetes, candidiasis

## Abstract

(1) Background: A clinical laboratory index to assess gut dysbiosis is the F/B ratio < 0.8. In fact, an elevated proportion of Firmicutes and a reduced population of Bacteroides in diabetes type 2 (T2D) subjects has been observed. This study aimed to detail the dysbiosis status in the Italian population, focusing on some pathogenic spectra (T2D) or metabolic disorders. (2) Material and methods: A quantity of 334 fecal samples was analyzed in order to perform genetic testing and sequencing. (3) Results: A trend in over imbalance was observed in the percentage of Proteobacteria (median value: 6.75%; interquartile range (IQR): 3.57–17.29%). A statistically significant association (χ^2^
*p* = 0.033) was observed between type of dysbiosis and T2D, corresponding to an Odds Ratio (OR) of 1.86. It was noted that females with cystitis/candidiasis are significantly prevalent in T2D patients (*p* < 0.01; OR: 3.59; 95% CI: 1.43–8.99). Although, in non-diabetic males, a sugar craving is significantly associated with the rate of dysbiosis in non-diabetic males (*p* < 0.05; OR 1.07; 95% CI 1.00–1.16). (4) Conclusion: In T2D patients, the Bacteroidetes/Firmicutes ratio was biased in favor of Proteobacteria, to be expected due to the nutritional habits of the patients. Thus, T2D females had altered gut permeability favoring the development of infections in the vaginal tract.

## 1. Introduction

The term “microbiota” identifies the symbiotic relationship of living microorganisms (Bacteria, Fungi, and Viruses) present in our body, defined by not only the community of the microorganisms but also their “theatre of activity” [1,2]. In fact, the microbiota interacts with several human organs, and its bacterial composition shows tissue-specific differences and influences the host health. One of the richest and largest of all human microbiota is the gut microbiota where millions of bacteria colonize the human intestine, especially the colon tract (70%). Here, the two dominant strains are Firmicutes and Bacteroides [3]. The adult microbiota remains stable over time. At the taxonomic level of Phylum, the composition between individuals is quite similar, while at the species (*spp*) taxonomic level, this similarity is lost. The microbiota at this level is quite a “signature” or a “footprint” to indicate a co-evolution of microorganisms with their hosts [4]. On the other hand, it is a quickly developing ecosystem in continuous interaction with the host body and external factors. Several causes influence and affect fluctuations in the composition of the microbiota—the genetic background of each person, even the type of delivery (C-section/natural birth), or environmental factors, such as diet, stress, smoking, lifestyle, drugs, encounters with other pathogens, age, and others [5,6].

The two predominant Phyla, Bacteroidetes and Firmicutes, represent 60–80% of the species in the gut microbiota. Another Phylum is Proteobacteria, including mainly Gram-negative, potentially pathogenic, and usually low-represented bacteria, such as Escherichia coli, Salmonella typhi, Vibrio cholerae, Helicobacter pylori, and Pseudomonas aeruginosa [7]. Another low-represented Phylum is Actinobacteria, which include Gram-positive bacteria, such as Bifidobacteria.

In a healthy individual, the microbiota’s metabolic activities and interactions influence the host state of health or disease. In fact, intestinal bacteria play a role in digestive function for vitamin synthesis (vitamin K, folic acid, vitamin B2, B3, B5, B6, B7, and B12) and for the digestion of dietary fibers [3]. However, the microbiota’s protective aspect is performed by the mutual interaction between the gut microbiota and the host immune system, which is crucial for maintaining homeostasis and in stimulating its response [4]. The continuous cooperation between the intestinal bacteria and the immune system is associated with the intestinal mucosa, namely the gut-associated lymphoid tissue (GALT), which is crucial for immune tolerance to commensals and to food antigens, as much as in maintaining efficiency in eliminating potentially harmful factors [8,9]. Moreover, the microbiota *spp* encode metabolic genes which regulate the degradation of polysaccharides (saccharolytic fermentation) and proteins (putrefactive fermentation) and the synthesis of short-chain fatty acids (SCFAs) [9]. Some useful and beneficial SCFAs are butyrate and propionate as regulators of intestinal physiology controlling anti-inflammatory and immunomodulatory functions [10]. Sequentially, diet can have a great impact on the microbiota [11], such as a Western-style diet which is characterized by a high intake of animal proteins, saturated fats, and simple sugars, and is low in fruit, vegetables, and other fibers [12]. Diets that are imbalanced by being low in complex polysaccharides (and therefore, based on carbohydrates) and rich in protein intake can change the composition of the gut microbiota by decreasing the percentages of Bacteroidetes (B) over Firmicutes (F), leading to irregular saccharolytic fermentation [13]. This imbalance in the microbial ecosystem is called dysbiosis, defined by the altered representation of Phyla percentages [14]. Dysbiosis can lead to the numerical expansion of some/certain potentially pathogenic commensal bacterial species. Their consequent dominance in the niche and symptomatology, such as the production of gas that is led by fermentation, could be toxic if caused by an increase in Firmicutes over Bacteroides in a dysbiosis status. Other potentially toxic products are generally immediately eliminated by secondary fermentation, performed by other bacterial consortia (reducing acetogenins, sulfate-reducing bacteria, and methanogens) [15]. Another scenario is the growth of Proteobacteria, consequent to a Bacteroidetes decrease, id est, a putrefactive fermentation, with a higher intake and protein degradation. Some potentially pathogenic species, which could promote the dysbiosis status, belong to the Proteobacteria Phylum [14]. In this context, a useful clinical laboratory index to establish the microbiota status and to diagnose the dysbiosis status is the F/B ratio < 0.8. Many studies reported an increase in this ratio in obese people compared with people with normal weight [16,17,18]. Furthermore, the variations of the other Phyla, mainly Proteobacteria and Actinobacteria, must also be related to the pathological changes. An elevated proportion of Firmicutes and a reduced population of Bacteroides were observed in obese and diabetes type 2 (T2D) subjects [1]. T2D is characterized by the decreased production of butyrate, one of the SCFAs that supports proper function of β-cells in the pancreas, especially after food intake. Butyrate contributes to the modulation of immune system functions and protection against pathogen invasion [10]. It also activates intestinal gluconeogenesis and, as a result, favorably affects glucose homeostasis. In this case, the F/B ratio is > 0.8 and it correlates with the status of saccharolytic fermentation. It has been confirmed that several species belonging to the Firmicutes Phylum have a higher and better degradation capacity of complex sugars and fatty acids; therefore, their increase contributes to the development of obesity [2,3,8].

A lower abundance of certain butyrate-producing bacteria, such as class Clostridia and genus Faecalibacterium (Phylum Firmicutes), have been observed in patients with T2D. Larsen et al. reported a lower abundance of Clostridia in patients with T2D, while two other studies [19] observed that patients with T2D had a lower abundance of Faecalibacterium, due to the adoption of a low-sugar diet. The same scenario was observed in obese patients following a low-calorie diet (low-sugar, fiber-rich, and high-protein intake) [20]. Modulating the gut microbiota with different diets (rich in fiber) and probiotic supplementation is a promising approach for the treatment and prevention of obesity and other intestinal inflammatory diseases [21]. In light of this data, the aim of this study was to monitor the microbiota population and heterogeneity in the Italian population. Knowing the troublesome implications of dysbiosis and the wide data available on it, this study has been focusing to detailing the gut microbiota status in the Italian population, drawing the attention on peculiar profiles that could reflect some pathogenic spectra or metabolic disorders, such as diabetes.

## 2. Materials and Methods

### 2.1. Population

This study used original data from the Lifebrain Cerba Healthcare laboratories—groups of private clinical structures in Italy. A quantity of 334 fecal samples was analyzed to perform a gut microbiota genetic test, accompanied by filling out a questionnaire in order to provide a medical history useful for the interpretation of the results obtained and to provide a correct diagnosis using a specific test kit for the stool sample collection. The samples were collected from people aged between 2 and 87 years old. For the analysis, the exclusion criteria adopted were: under 16 and over 80 years old; patients misidentified. In conclusion, a total of 314 samples were selected. Informed consent was obtained from all the subjects enrolled in the study. This study was approved by the Ethics Committee of Palermo 1, Italy (Protocol no. 8 14/09/2022, Identifier, NCT05565651). The study has been conducted following the Declaration of Helsinki (2013).

### 2.2. Fecal DNA Extraction and 16s rRNA Gene Sequencing

Microbial DNA was extracted from a 200 µL fecal sample using a MagNA Pure Compact Nucleic Acid Isolation Kit (Roche Diagnostic, Switzerland). The nucleic acid isolation procedure is based on magnetic-bead technology. Briefly, a MagNA Pure Compact nucleic acid isolation procedure follows the subsequent steps. The samples are lysed by incubation with Proteinase K and a special lysis buffer containing a chaotropic salt. Secondly, Magnetic Glass Particles (MGPs) are added, and nucleic acids are immobilized on the MGP surfaces; unlikely unbound substances are removed by several washing steps. Finally, purified DNA is eluted from the MGPs and can be used for downstream assays.

PCR amplification was conducted using a 338F forward primer 5′-ACTCCTACGG GAGGCAGCAG-3′ and an 806R reverse primer 5′-GGAC TACHVGGGTWTCTAAT-3′. PCR was run using the following program: 95 °C for 3 min, followed by 21 cycles of 95 °C for 30 s, 56 °C for 30 s, and 72 °C for 30 s, with a final extension at 72 °C for 5 min.

Next-generation sequencing (NGS) of the bacteria-specific 16S ribosome gene was performed utilizing a microbiota solution B kit—hypervariable regions V3-V4-V6 (Arrow Diagnostics S.r.l., Genoa, Italy). The B kit was composed of Enzyme Mix 1 solution containing the enzyme mixture for the PCR target, Enzyme Mix 2 solution containing the enzyme mixture for the PCR index, Amp Mix V3-V6 solution of degenerated oligonucleotides for amplifying hypervariable regions V3-V4-V6 of the bacterial 16S rDNA gene, and an oligonucleotide solution for indexing amplified samples with the PCR target (Index Mix). An Illumina^®^ MiSeq ™ 6000 system platform (Illumina Inc., San Diego, CA, USA) was used for sequencing.

### 2.3. Sequencing Data Analysis

Raw sequencing data were uploaded to a bioinformatic system. Taxonomic assignment and a bioinformatic analysis were performed using the MicrobAT^®^ software (Microbiota Analysis Tool; SmartSeq S.r.l., Novara, Italy) [22,23]. In the first phase of the analysis, reads were cleaned by a dedicated algorithm to remove short, low-quality sequences. Taxonomic assignment was then made by aligning the remaining sequences with the Lifebrain Cerba Healthcare laboratory reference database.

The data extrapolated from the data analysis showed the following parameters: dysbiosis index, microbial heterogeneity, F/B ratio, and percentages of Bacteria Phyla and pathogens compared with our reference population.

### 2.4. Statistical Analysis

In order to apply a parametric analysis, normality was assessed with the Shapiro–Wilks test. Descriptive statistics were represented as mean and standard deviation (SD), and the statistical significance of the difference between genders was assessed with ANOVA.

To assess the abundance and richness of bacteria, a multiple measurement ANOVA was performed, and the Sphericity (ε; the equality of variances of the differences between measurements) was assessed. If Mauchly’s Sphericity test returned a small *p*-value, it was applied a correction to the index of non-sphericity: when ε ≤ 0.75, the Greenhouse–Geisser correction was used; when ε > 0.75, the Huynh–Feldt correction was used. Once the corrections were applied, the data were represented as median and Interquartile range (IQR).

The association between nominal data was performed with the Chi-Square test or exact test (Fisher–Freeman–Halton) according to whether the counts in at least one cell of the contingency table were < 5. In stratified tables, where strata were used to control for a specific factor, the Odd Ratio (OR) was computed according to the procedure of Mantel–Haenszel. In the case of multiple simultaneous comparisons of the same sample, the Bonferroni correction was applied to the level of statistical significance to compensate for the inflation of a type I error. Data were analyzed with StatsDirect 2.7.2 (StatsDIrect Ltd., Wirral, UK) and with MedCalc (MedCalc Software Ltd., Ostend, Belgium). It was assumed a 95% odd Confidence Interval (CI).

## 3. Results

A quantity of 334 fecal samples was analyzed to assess the gut microbiota composition and to determine the dysbiosis status level of each patient who requested the analysis. The samples processed for this study totaled 314: 89 samples from males, and 225 from females.

### 3.1. Epidemiological Data

Table 1 reports anamnestic data, anthropometric measurements, and the presence of diseases (such as cardiovascular disease—CVD, diabetes, autoimmune disorders, celiac disease, and thyroid disorders) in the investigated population. No significant difference between genders was observed.

### 3.2. Microbiota Heterogeneity

Figure 1A,B shows the total microbiota heterogeneity in the Italian population tested. As can be observed, most of the patients had limited heterogeneity, indicating low biodiversity. The biodiversity was assessed using an index, defined by the MicrobAT^®^ software (Microbiota Analysis Tool; SmartSeq S.r.l., Italy) [22,23]. Biodiversity is defined by the number of *spp* populations observed and is an indicator of the health of the microbiota—the lower the result is, the lower the number of microbial populations detected. Based on the biodiversity and the symptoms detected by the questionnaire, the present work established an index of biodiversity:-Under 200: very low biodiversity-200–400: low biodiversity-400–600: mild biodiversity-600–800: high biodiversity-Over 800: healthy biodiversity

The widow in Figure 1A, illustrates levels of biodiversity from 85.7 to 757.2. The results indicate a strong presence in the low (mean: 306.06, number (n): 201) and mild range (mean: 472.00; number: 70) of biodiversity, while high biodiversity is a low-represented range (mean: 706.87, n: 16) similar to the very low biodiversity range (mean: 176.60, n: 24). A healthy biodiversity range is registered in just three cases (mean: 831.669).

Figure 1B, showed the percentages of the most important Phyla in dysbiosis patients (291 patients); eubiosis was excluded in the plot (23 patients). The data reported a trend in over imbalance in the percentage of Proteobacteria (median value: 6.75%; interquartile range or IQR: 3.57–17.29%) that doubled their median value despite their health reference value. Another sign of dysbiosis was represented by the lower percentages in Firmicutes compared with their health reference value (median value: 37.63%; IQR: 30.49–44.91%) and in Bacteroidetes (median value: 45.78%; IQR: 39.71–50.83%). The Actinobacteria agreed with the healthy state (median value: 0.99%; IQR: 0.25–2.80%). The health reference percentages are stated in Table 2.

### 3.3. Microbiota Abundance and Richness

The microbiota abundance and richness assessments were carried out employing the overall samples, the eubiosis status, and the dysbiosis statuses. These parts were represented as cumulative frequency due to the distribution of our large-grouped data sets. The sphericities were determined and used to calculate the statistical parameters such as median, IQR, 95% CI, and minimum and maximum value of the percentages. These parameters were reported in Table 3.

Cumulative frequency curves were shown in Figure 2. The assessed Pairwise comparisons had *p* > 0.001 for each graph. In the overall samples and in the eubiosis cases, the cumulative frequency curves have a distinctive ‘s-shape’ for the Bacteroidetes (median value: 45.42% and 42.18%, respectively) and Firmicutes (37.45% and 38.11%, respectively). Meanwhile, the Proteobacteria (median value: 7.05% and 4.74%, respectively) and the Actinobacteria (median value: 1.01% and 0.78%, respectively) were in the range of 2–5% and 0–1%. In the mild dysbiosis status, the graph showed not quite s-shaped curves and a wider range of the Bacteroidetes (median value: 47.83%) and Firmicute percentages (median value: 41.89%). Similarly, the moderate dysbiosis status cases revealed a decrease in Firmicute percentages (median value: 39.39%), with a slight increase in the Proteobacteria percentage (median value: 6.12%). The severe dysbiosis status profile witnessed a further decrease in the percentages of Bacteroidetes and Firmicutes (median value: 46.24% and 32.36%, respectively) and a strong increase in Proteobacteria (median value: 12.45%). This trend is even more evident in the graph of the critical dysbiosis status where the lines representing percentages of Bacteroidetes (median value: 39.12%), Firmicutes (median value: 23.43%), and Proteobacteria (35.67) are slightly overlapped. In addition, in this last graph, the Actinobacteria percentages were half their normal values (median value: 0.54%).

### 3.4. Dysbiosis and Pathologies

From Table 1, many patients with dysbiosis have pathologies such as CVD (n. 90) and T2D (n. 91). For this reason, we decided to perform a deeper investigation into the type and the rate of dysbiosis in these subjects. CVD patients did not show any significant difference regarding dysbiosis and the presence of CVD, while we observed a correlation between dysbiosis status and diabetic pathology. Figure 3 highlights the significant presence and the type (putrefactive or fermentative) of dysbiosis in T2D (*p* < 0.05), despite euglycemic subjects. The probability (OR) of developing putrefactive dysbiosis is 1.86 times greater than that of developing fermentative dysbiosis in T2D patients.

At this point, considering this statistically significant difference described above, we decided to examine if some symptoms were linked to dysbiosis in the euglycemic and T2D patients, through a risk model (logistic regression) analysis. 

Figure 4A, shows that females with cystitis/candidiasis are significantly prevalent in the T2D population (*p* < 0.01; OR: 3.59; 95% CI: 1.43–8.99). Moreover, they are characterized by severe dysbiosis.

In non-diabetic males, a sugar craving is significantly associated with the rate of dysbiosis in non-diabetic males (*p* < 0.05; OR 1.07; 95% CI 1.00–1.16). However, in comparing sugar cravings between males and females, the probability (OR), based on BMI and age, of causing a sugar craving is 2.13 (95% CI 1.04–4.37). 

## 4. Discussion

In the last 15 years, the gut microbiota composition and its dysbiosis have been associated with various diseases, and ever more data are shown in the literature [1,14,24,25], as many studies have focused on finding a correlation between dysbiosis status and the development and evolution of a disease in order to try to answer the “who comes first” dilemma. 

The overall data analyzed in this study confirmed the sophisticated processes of microbiota dysbiosis. Was observed an increase in Proteobacteria of 63% and a decrease in the sum of Bacteroidetes (−15%) and Firmicutes (−13%), regardless of gender. A eubiotic microbiota has a high biodiversity (over 600 *spp* and up to 1000) which allows for a balance between the various species and its positive action in fermentation and in the various roles mentioned above. Meanwhile, it was noted a dysbiosis status in all patients who declared symptoms (bloating, sugar craving, headache, diarrhea, constipation) and pathologies/infections (CVD, Diabetes, Autoimmune disorders, Candidiasis) in the questionnaire. Furthermore, Figure 2 confirmed this trend. Thus, the middle values of the bacterial percentages decreased between the dysbiosis statuses despite the eubiosis, and in the several and the critical statuses of dysbiosis as well, as was shown by the increase in Proteobacteria percentages that are overlapping with the Bacteroidetes and Firmicutes percentages. This reflects biodiversity species poorness. 

From our findings, the gut microbiota composition in patients with T2D is different from that in healthy subjects, supporting our hypothesis that gut alterations and dysbiosis status are implicated in pathological disorders. However, the analysis carried out on patients suffering from T2D with insulin resistance showed discordant values from the literature [19]. The dysbiotic state of T2D patients is usually associated with an important increase in Firmicutes at the expense of Bacteroidetes (F/B > 0.8) which causes an increase in caloric absorption; Firmicutes is the Phylum with the greatest number of *spp* and which carries out saccharolytic fermentation starting from the degradation of complex sugars. Typically, this causes an increase in the release of gas and in the permeability of the gastroepithelial barrier, which leads to weight gain, inflammation, discomfort, and swelling [26]. On the other hand, the main adjustment seen in the F/B ratio and the percentages of Proteobacteria is in agreement with precedent research, where an increase has been noticed in the number of opportunistic pathogens and a decrease in bacteria-producing butyrate, one of the most important SCFAs [27]. One of the first studies on the gut microbiota composition in subjects with T2D, conducted by Larsen et al., showed decreased levels of the Phylum Firmicutes and of the class Clostridia. Instead, in this work, the Bacteroidetes/Firmicutes ratio was biased in favor of Proteobacteria. The data analysis showing the anachronistic increase in Proteobacteria is likely due to the diet followed by the patients—decreased complex carbohydrate intake and a more protein-based diet, which is the key to proteolytic fermentation. This is also suggested by the non-significant data that did not report obese or severe overweight people among the ones who performed the diagnostic test. Furthermore, studies indicate a potentially beneficial effect of Bacteroides on glucose metabolism and suggest an explanation for the negative correlation between Bacteroides and T2D, as observed in this study.

Furthermore, confirming diet influence in modulating the gut and its balance, people who declared following a plant-based diet showed a non-significant status level of dysbiosis classified as moderate or light [28,29,30,31], whereas the results regarding diabetic people showed that it is essential to follow a balanced diet, possibly low in fatty acids and high in complex polysaccharides, and with a correct intake in proteins. In fact, the latter, if overloaded could trigger putrefactive fermentation causing alterations in the intestinal tract, such as increased inflammation, gas release, abdominal pain, and an increase in gut permeability [7].

Moreover, the gut microbiota of individuals suffering from T2D is characterized by an increase in opportunistic pathogens (such as Fungi and Proteobacteria *spp*), in agreement with previous studies [25,32]. This altered status can increase the gut permeability, thus affecting the defensive barrier action of the microbiota and permitting some *spp* to spread in the bloodstream favoring the development of even life-threatening infections in other organs [33,34]. The results obtained about the significant correlations between T2D and cystitis/candidiasis in women confirm that dysbiosis status can cause an imbalance in the fungi populations which cohabit our gut, and in their proliferation. Candida albicans can reach the vagina and the urinal tract [32,35] through the bloodstream whereby implementing the onset of candidiasis and cystitis, respectively, as declared by patients. In the future, the zonulin level (dysbiosis marker) [36] should be investigated in order to evaluate its influence on gut permeability, such as investigating possible hormonal imbalances.

It is noteworthy that the data obtained regarding the symptom “sugar craving” show that it is more present in euglycemic men than in women. Microbial communities with lower biodiversity could lead to an overgrowth by one or more species, whereby providing an increased ability to manufacture behavior-altering neurochemicals and hormones, leading to an increased craving for high-fat and high-carbohydrate diets, as discussed in the literature [37]. It would be interesting to confirm the possible correlation between gender–biodiversity–sugar craving (gender-biased effect) shown by our data, by amplifying the male population. In addition, the sugar craving has not been measured quantitatively and it could be another biased source. However, these results correlate with dysbiosis status levels in that those with severe or critical status levels of dysbiosis showed a more significant correlation, as expected. Moreover, the next step of our study would be to consider performing an analysis at the genus and species levels.

## 5. Conclusions

Overall, given the personal and societal impact on diabetes, our data support that dysbiosis was confirmed in T2D patients, due to the Bacteroidetes/Firmicutes ratio being biased in favor of Proteobacteria, and likely based on the T2D patients’ nutritional habits as well. For this, an in-depth study will be required to fully assess the bacterial profile and whether microbiome modulation through probiotics may be a helpful alternative approach to improve insulin control and reduce the risk of T2D complications.

## Figures and Tables

**Figure 1 ijerph-19-15913-f001:**
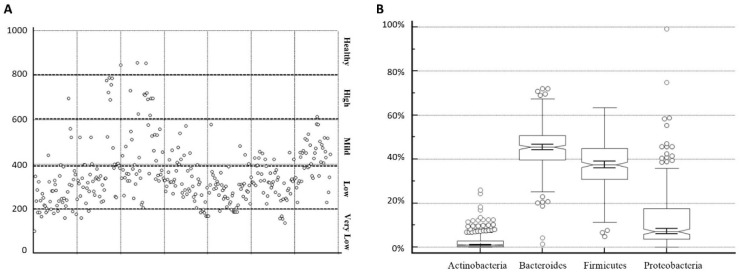
Panel (**A**) Levels of biodiversity in the population tested, divided by the index established. Panel (**B**) Whisky box representation of the *spp* percentages detected.

**Figure 2 ijerph-19-15913-f002:**
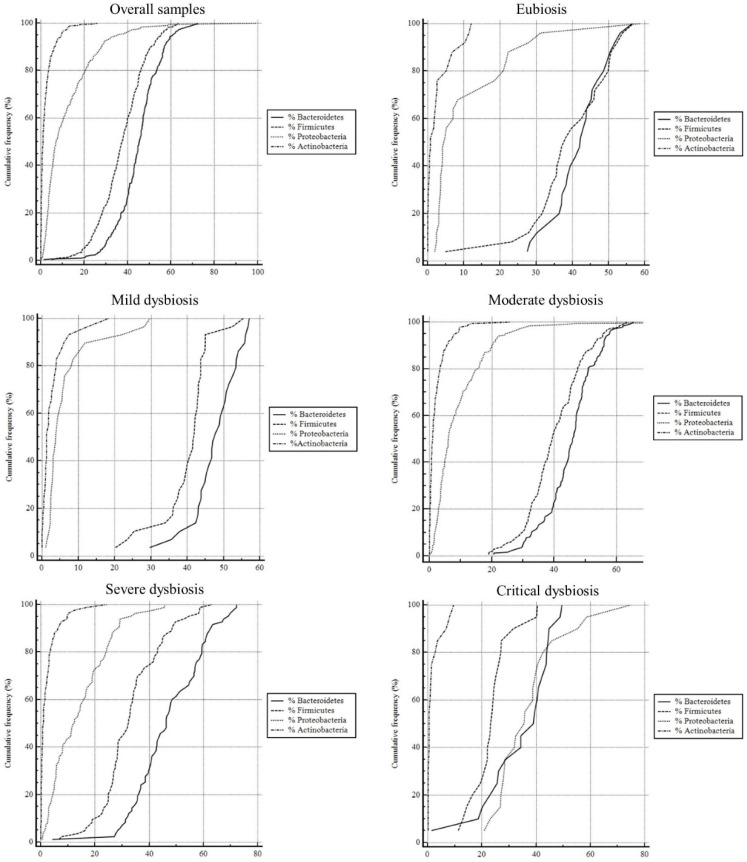
Cumulative frequencies of the Phyla observed for all the statuses. The *x*-axis is marked with the class intervals from the data set on a continuous scale. The *y*-axis represents the quartiles of the cumulative frequencies.

**Figure 3 ijerph-19-15913-f003:**
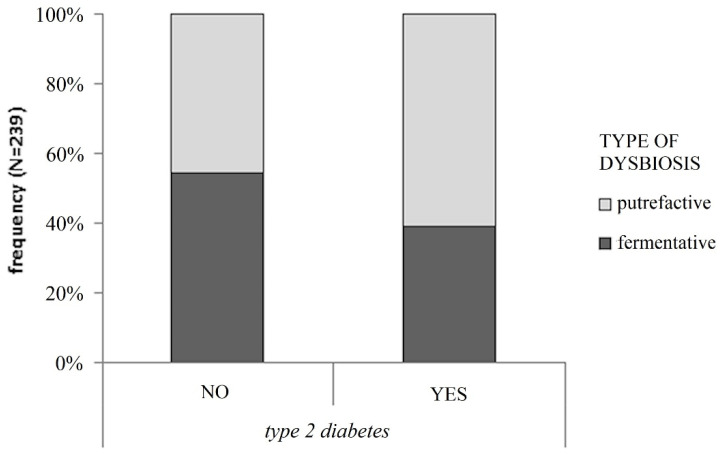
Statistically significant association (χ^2^
*p* = 0.033) between type of dysbiosis and type 2 diabetes regardless of gender, which corresponds to an Odds Ratio of 1.86 (95% CI: 1.05–3.29).

**Figure 4 ijerph-19-15913-f004:**
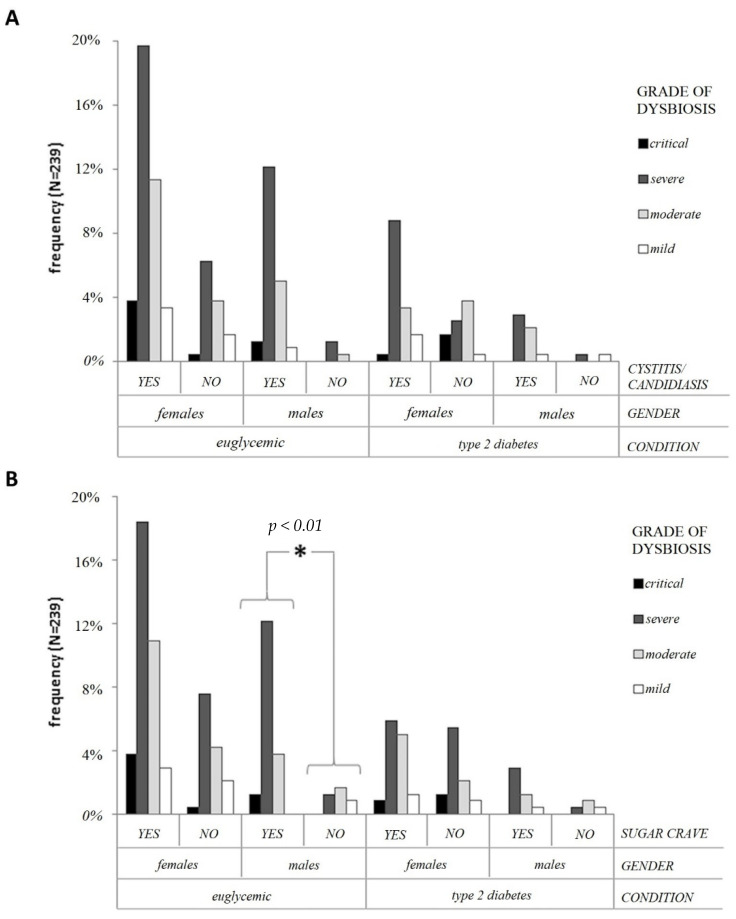
Panel (**A**) Association of cystitis/candidiasis in subjects with a grade of dysbiosis, controlled for gender and type 2 diabetes. Exact statistical significance was adopted. Panel (**B**) Association of sugar cravings in subjects with a grade of dysbiosis, controlled for gender and type 2 diabetes. The star marks the statistically different distribution of counts between non-diabetic males that show or do not show sugar cravings. *p* < 0.01 refers to exact statistical significance.

**Table 1 ijerph-19-15913-t001:** Epidemiological findings of the study population (BMI—body mass index; CVD—cardiovascular disease; N—number; SD—standard deviation; and y.o.—years old).

Descriptive Statistics			
	Females	Males	Total
N	225	89	314
AGE y.o. mean (SD) min–max	46.4 (15.0) 16–77 y.o.	49.0 (16.7) 17–80 y.o.	n/a
BMI mean (SD)	23.1 (4.4)	24.3 (3.3)	n/a
CVD	71% (n. 64)	29% (n. 26)	90
Diabetes, type 2	77% (n. 70)	23% (n. 21)	91
Autoimmune disorders	76% (n. 26)	29% (n. 8)	34
Celiac disease	92% (n. 12)	8% (n. 1)	13
Thyroid disorders	76% (n. 19)	24% (n. 16)	25

**Table 2 ijerph-19-15913-t002:** Median values of the percentages of the *spp* detected compared with the health reference values.

*spp*	Health Reference Values	Median Values Obtained
Bacteroidetes	50–55%	45.78% (lower)
Firmicutes	40–45%	37.63% (lower)
Proteobacteria	2–3%	6.75% (higher)
Actinobacteria	1%	0.99% (same)

**Table 3 ijerph-19-15913-t003:** Median values, IQR, 95% CI, and minimum and maximum value of the percentages of the Phyla detected for all the statuses observed.

Overall Samples	% Bacteroidetes	% Firmicutes	% Proteobacteria	% Actinobacteria
(*n* = 314)				
*Median*	45.42	37.45	7.05	1.01
*IQR (25th–75th quartile)*	39.63 to 50.70	30.87 to 44.87	3.62 to 17.44	0.27 to 2.80
*Minimum*	1.34	4.88	0	0
*Maximum*	72.12	63.3	99.28	25.94
*95% CI*	43.96 to 46.12	36.28 to 38.56	10.89 to 13.57	1.94 to 2.67
**Eubiosis**	**% Bacteroidetes**	**% Firmicutes**	**% Proteobacteria**	**% Actinobacteria**
**(*n* = 26)**				
*Median*	42.18	38.11	4.74	0.78
*IQR (25th–75th quartile)*	37.05 to 46.85	33.55 to 48.20	3.42 to 18.38	0.17 to 2.54
*Minimum*	27.48	4.88	1.89	0
*Maximum*	56.53	56.78	58.68	11.99
*95% CI*	38.82 to 45.09	34.51 to 43.92	5.97 to 16.89	1.17 to 4.24
**Mild dysbiosis**	**% Bacteroidetes**	**% Firmicutes**	**% Proteobacteria**	**% Actinobacteria**
**(*n* = 30)**				
*Median*	47.83	41.89	3.76	1.34
*IQR (25th–75th quartile)*	43.88 to 52.50	37.72 to 43.63	2.53 to 6.29	0.79 to 3.43
*Minimum*	29.73	20.27	1.01	0
*Maximum*	57.1	55.37	29.71	18.41
*95% CI*	45.22 to 50.12	37.32 to 42.91	3.86 to 9.52	1.39 to 4.44
**Moderate dysbiosis**	**% Bacteroidetes**	**% Firmicutes**	**% Proteobacteria**	**% Actinobacteria**
**(*n* = 184)**				
*Median*	45.99	39.39	6.12	1.04
*IQR (25th–75th quartile)*	40.60 to 50.02	34.24 to 46.29	3.56 to 13.43	0.28 to 2.76
*Minimum*	20.58	18.99	0	0
*Maximum*	65.52	63.3	99.28	25.94
*95% CI*	43.99 to 46.44	38.82 to 41.51	7.97 to 11.01	1.68 to 2.61
**Severe dysbiosis**	**% Bacteroidetes**	**% Firmicutes**	**% Proteobacteria**	**% Actinobacteria**
**(*n* = 84)**				
*Median*	46.24	32.36	12.45	0.97
*IQR (25th–75th quartile)*	38.45 to 57.24	26.56 to 40.68	5.33 to 21.87	0.25 to 3.02
*Minimum*	4.27	6.57	0.7	0
*Maximum*	72.12	63.07	45.76	24.25
*95% CI*	44.42 to 50.01	30.76 to 35.68	11.97 to 16.67	1.57 to 3.28
**Critical dysbiosis**	**% Bacteroidetes**	**% Firmicutes**	**% Proteobacteria**	**% Actinobacteria**
**(*n* = 21)**				
*Median*	39.12	23.43	35.67	0.54
*IQR (25th–75th quartile)*	26.01 to 43.67	20.42 to 26.30	28.06 to 40.97	0.21 to 1.76
*Minimum*	1.34	11.31	20.81	0.15
*Maximum*	49.49	40.39	74.79	9.5
*95% CI*	28.68 to 40.13	20.18 to 27.25	31.41 to 43.74	0.59 to 3.25

## Data Availability

Not applicable.

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
