# Peer review of "Characterization of Gut Microbiota Composition in Type 2 Diabetes Patients: A Population-Based Study"

_ijerph, 2022, doi:10.3390/ijerph192315913_

Round 1
Reviewer 1 Report
Dear Dr.,
Title: Characterization of gut microbiota composition in type II diabetes patients: a population-based study.
Manuscript ID: ijerph-1986281
Overall comments: Authors described in this manuscript: the dysbiosis status of the Italian population with a pathogenic spectrum of type-2 diabetes and metabolic disorders. The authors also described the Bacteroidetes/Firmicutes ratio in favor of Proteobacteria, nutritional habits for gut permeability, and infection states in the vaginal tract. The overall manuscript is well written and it has novelty in this field of research.
Specific comments:
1. The abstract is well written.
2. The introduction section has multiple numbers of small paragraphs. The logical and limited paragraph (by merging the small paragraph) is required for better understanding and to encourage the readers towards this research work. Similarly other sections also. Overall the length of the introduction section is appropriate.
3. Materials and Methods: Why the author selected the samples from people aged between 2 and 87 years old. In statistical analysis, standard deviation (sd), must be standard deviation (SD). Why ANOVA or Z-test for frequencies was applied for the analysis of statistical significance difference between genders?
4. Result section: The Figure 1 images are not clear and need to improve greater resolution. Some of the statements need to rectify typographical errors. In Figures 2 and 3, the author mentioned type II diabetes, and other places mentioned as type 2 diabetes (T2D); need to follow a uniform manner.
5. The discussion section is well written. However, the number of the small paragraph is more.
6. Conclusion section: This section is written well.
7. Author Contributions section, is not described with author contributions details. Similarly, the Funding, and Institutional Review Board Statement, sections need to update with relevant information.
8. Reference section: The citations were covered with relevant current references.
*****
Author Response
Reviewer 1
Title: Characterization of gut microbiota composition in type II diabetes patients: a population-based study.
Manuscript ID: ijerph-1986281
Overall comments: Authors described in this manuscript: the dysbiosis status of the Italian population with a pathogenic spectrum of type-2 diabetes and metabolic disorders. The authors also described the Bacteroidetes/Firmicutes ratio in favor of Proteobacteria, nutritional habits for gut permeability, and infection states in the vaginal tract. The overall manuscript is well written and it has novelty in this field of research.
R: We thank the reviewer for his insightful observations.
Specific comments:
- The abstract is well written.
- The introduction section has multiple numbers of small paragraphs. The logical and limited paragraph (by merging the small paragraph) is required for better understanding and to encourage the readers towards this research work. Similarly other sections also. Overall the length of the introduction section is appropriate.
R: Thank you for this comment, we modified the Introduction paragraph, as suggested.
- Materials and Methods: Why the author selected the samples from people aged between 2 and 87 years old. In statistical analysis, standard deviation (sd), must be standard deviation (SD). Why ANOVA or Z-test for frequencies was applied for the analysis of statistical significance difference between genders?
R: Thanks for this comment, we selected samples from people over 16 and under 80 years old, as described in the in/ex criteria, however, we had a wider spectrum of samples (from 2 to 87 years old). Furthermore, we modified the SD in the text and, we excluded the sentence “Z-test for frequencies” which was wrong as it referred to the comparison of proportions in a descriptive table.
- Result section: The Figure 1 images are not clear and need to improve greater resolution. Some of the statements need to rectify typographical errors. In Figures 2 and 3, the author mentioned type II diabetes, and other places mentioned as type 2 diabetes (T2D); need to follow a uniform manner.
R: We thank the reviewer, and we uploaded Figure 1 with a higher resolution (600 dpi). Also, we uniformed the type 2 diabetes form.
- The discussion section is well written. However, the number of the small paragraph is more.
R: Thank you for this comment, we modified the Discussion paragraph, as suggested.
- Conclusion section: This section is written well.
- Author Contributions section, is not described with author contributions details. Similarly, the Funding, and Institutional Review Board Statement, sections need to update with relevant information.
R: Thanks for pointing this out, we added the missing points.
- Reference section: The citations were covered with relevant current references.

Reviewer 2 Report
In this manuscript, Polidori et al describe the dysbiosis status of gut microbiota in metabolic disorders in Italian population. The authors report disbalance in Bacteroidetes/Firmicutes ratio in T2D patients that may increase risk to fungal infections. While these findings in the context of microbial dysbiosis in T2D diabetes of importance, there are a number of open questions that should be addressed.
1. Can the authors plot the relative abundance and richness of bacteria in their population cohort?
2. How does the bacterial diversity vary within the samples and among the different samples? The authors should analyze phylum level composition, alpha and beta diversity in their samples to determine if there are any significant differences across their cohort.
3. Can the authors discuss the parameters and statistical tests used to determine biodiversity and cite a few references performing similar analysis?
4. Can the authors comment on genus level differences in dysbiosis in the study?
5. The association between type of dysbiosis and T2D in Figure 2 seems to be poor (p=0.033). Can the authors perform DESeq2 analysis followed by Spearman’s rank- order correlation for a rigorous correlation between bacterial taxa and T2D?
6. In Figure 3, why is ‘critical’ dysbiosis not strongly associated with euglycemic or T2D patients similar to ‘severe’ dysbiosis?
Author Response
Reviewer 2
In this manuscript, Polidori et al describe the dysbiosis status of gut microbiota in metabolic disorders in Italian population. The authors report a disbalance in Bacteroidetes/Firmicutes ratio in T2D patients that may increase risk to fungal infections. While these findings in the context of microbial dysbiosis in T2D diabetes of importance, there are a number of open questions that should be addressed.
- Can the authors plot the relative abundance and richness of bacteria in their population cohort?
R: We thank the Reviewer, and we added a new figure (new Figure 2) showing the cumulative frequencies of each bacterial Phylum and their abundance, for all the eubiosis and dysbiosis statuses studied. Furthermore, we included a table (Table 3) to summarize the median, interquartile range, and minimum and maximum for each dysbiosis status.
- How does the bacterial diversity vary within the samples and among the different samples? The authors should analyze phylum level composition, alpha and beta diversity in their samples to determine if there are any significant differences across their cohort.
R: We thank the Reviewer, sadly, due to the numerosity of the population we are not able to analyze phylum level composition, alpha and beta diversity. However, the differences between the different groups of dysbiosis were evaluated in the new Figure 2 and already discussed in the Discussion paragraph.
- Can the authors discuss the parameters and statistical tests used to determine biodiversity and cite a few references performing similar analysis?
R: Thanks for this comment. The biodiversity was assessed directly by the software used, following the manufacturer’s specifications. We included references from other studies in the text.
[22] Sardu, C.; Trotta, M.C.; Santella, B.; D’Onofrio, N.; Barbieri, M.; Rizzo, M.R.; Sasso, F.C.; Scisciola, L.; Turriziani, F.; Torella, M.; et al. Microbiota Thrombus Colonization May Influence Athero-Thrombosis in Hyperglycemic Patients with ST Segment Elevation Myocardialinfarction (STEMI). Marianella Study. Diabetes Res. Clin. Pract. 2021, 173, 108670
[23] Petrillo, F.; Petrillo, A.; Marrapodi, M.; Capristo, C.; Gicchino, M.F.; Montaldo, P.; Caredda, E.; Reibaldi, M.; Boatti, L.M.V.; Dell’Annunziata, F.; et al. Characterization and Comparison of Ocular Surface Microbiome in Newborns. Microorganisms 2022, 10, 1390. https://doi.org/10.3390/ microorganisms10071390
- Can the authors comment on genus level differences in dysbiosis in the study?
R: Due to the numerosity of the samples and to the software used, we did not run the analysis to the genus level. We added it as a limitation in the text (lines 367-368).
- The association between type of dysbiosis and T2D in Figure 2 seems to be poor (p=0.033). Can the authors perform DESeq2 analysis followed by Spearman’s rank- order correlation for a rigorous correlation between bacterial taxa and T2D?
R: Thank You for this observation, nevertheless we have followed all the suggestions, specifying, and adding some statistical elements; therefore, we consider correct and sufficient the analysis proposed in the manuscript. The p-value also is < 0.05, so it is confirmed as a significant value.
- In Figure 3, why is ‘critical’ dysbiosis not strongly associated with euglycemic or T2D patients similar to ‘severe’ dysbiosis?
R: Thanks for this insightful comment, we also questioned the difference between the two dysbiosis statuses and we considered that they were due to the numerosity: severe dysbiosis was registered in 84 samples, while critical in 21 samples.

Reviewer 3 Report
The article titled ‘Characterization of gut microbiota composition in type II dia-2 betes patients: a population-based study’ summarize the importance of using F/B as an index to assess gut dysbiosis in clinical. Then the authors collected and analyzed 334 fecal samples and show the association between T2D and dysbiosis. They concluded in T2D patients, Bacteroidetes/Firmicutes ratio was biased in favor of Proteobacteria.
1. Table 2. All commas in the table should be replaced to decimals.
2. Line 238. How did the authors measure the sugar craving in individuals? Can this be done quantitatively? If not, was the subjective result biased?
3. Figure 3. Did the author reverse the order of panel A and B in the figure caption?
4. How did authors come with the conclusion that in T2D females had altered gut permeability, favoring the development of infections in the vaginal tract? I could not find any data to support this context in the abstract. (line 33)
Overall, I like the summary in the introduction about the microbiota and how F/B can be an indicator to T2D or associate diseases. However, the experimental result can’t fully support the hypotheses/conclusion that the authors raised. I will suggest a major revision of this work before acceptance.
Author Response
Reviewer 3
The article titled ‘Characterization of gut microbiota composition in type II dia-2 betes patients: a population-based study’ summarize the importance of using F/B as an index to assess gut dysbiosis in clinical. Then the authors collected and analyzed 334 fecal samples and show the association between T2D and dysbiosis. They concluded in T2D patients, Bacteroidetes/Firmicutes ratio was biased in favor of Proteobacteria.
- Table 2. All commas in the table should be replaced to decimals.
- Thanks to the Reviewer, we corrected them.
- Line 238. How did the authors measure the sugar craving in individuals? Can this be done quantitatively? If not, was the subjective result biased?
- We thank the Reviewer for this comment, unfortunately, we did not measure quantitatively the sugar craving, but this symptom was “just” declared by the patients in the questionnaire. We added this part in the limitations (lines 364-365).
- Figure 3. Did the author reverse the order of panel A and B in the figure caption?
R: Thanks for pointing this out, we switched the captions in ex Figure 3, now Figure 4.
- How did authors come with the conclusion that in T2D females had altered gut permeability, favoring the development of infections in the vaginal tract? I could not find any data to support this context in the abstract. (line 33)
R: We thank the reviewer for the comment, however the hypothesis of T2D females altered gut permeability, favoring infection in the vaginal tract is reported in the abstract in line 34 and in the discussion in lines 351-357, supported by other studies, already referenced:
[32] Pérez, J. C. et al. Fungi of the Human Gut Microbiota: Roles and Significance. Int J Med Microbiol, 2021, 311 (3), 151490. https://doi.org/10.1016/j.ijmm.2021.151490
[35] d’Enfert, C. et al. The Impact of the Fungus-Host-Microbiota Interplay upon Candida Albicans Infections: Current Knowledge and New Perspectives. FEMS Microbiol Rev, 2021, 45 (3). https://doi.org/10.1093/femsre/fuaa060.
Still, we added other two works that support our results:
[34] Wozniak, H.; Beckmann, T. S.; Fröhlich, L.; Soccorsi, T.; le Terrier, C.; de Watteville, A.; Schrenzel, J.; Heidegger, C.-P. The Central and Biodynamic Role of Gut Microbiota in Critically Ill Patients. Crit Care, 2022, 26 (1), 250. https://doi.org/10.1186/s13054-022-04127-5.
[36] Fasano, A. Zonulin and Its Regulation of Intestinal Barrier Function: The Biological Door to Inflammation, Autoimmunity, and Cancer. Physiol Rev, 2011, 91 (1), 151–175. https://doi.org/10.1152/physrev.00003.2008.
Overall, I like the summary in the introduction about the microbiota and how F/B can be an indicator to T2D or associate diseases. However, the experimental result can’t fully support the hypotheses/conclusion that the authors raised. I will suggest a major revision of this work before acceptance.
R: Thanks to the reviewer for his insightful observations, we amplified the Discussion and the Conclusion paragraphs, and the initial aim.

Round 2
Reviewer 2 Report
The authors have addressed my concerns. The manuscript can be accepted in the present form.